# TimeCAT: Hierarchical Context-Aware Transformer with Dynamic Grouping for Time Series Forecasting

## Abstract

**Abstract**

Transformer-based models have achieved significant success in time series forecasting by modeling global dependencies through self-attention mechanisms. However, these models often rely on fixed patch settings with locality constraints, tokenizing time series into spatially connected sub-series. This approach can hinder the capture of semantic relationships and lead to computational inefficiencies, especially when dealing with long sequences with complex temporal dependencies. In this work, we introduce **TimeCAT**—a Time series Context-Aware Transformer that dynamically groups input sequences into semantically coherent groups, enabling efficient modeling of both local and global dependencies. By appending group and global tokens, TimeCAT facilitates fine-grained information exchange through a novel *Context-Aware Mixing Block*, which utilizes self-attention and MLP mixing operations. This hierarchical approach efficiently models long sequences by processing inputs in structured contexts, reducing computational overhead without sacrificing accuracy. Experiments on several challenging real-world datasets demonstrate that TimeCAT achieves consistent state-of-the-art performance, significantly improving forecasting accuracy and computational efficiency over existing methods. This advancement enhances the Transformer family with improved performance, generalization ability, and better utilization of sequence information.

## 1 Introduction

Time series forecasting plays a pivotal role in various domains such as finance (Zhang et al., 1998), weather prediction (Rasp & Lerch, 2018), energy management (Ahmed & Khalid, 2019), and healthcare (Cheng et al., 2017). Accurate long-horizon forecasting is essential for informed decision-making and strategic planning in these fields. The advent of deep learning has spurred significant advancements in modeling complex temporal patterns, with Transformer-based models emerging as a powerful tool due to their ability to capture long-range dependencies through self-attention mechanisms (Vaswani, 2017; Zhou et al., 2022a; Wu et al., 2023; Liu et al., 2024).

Despite their success, existing Transformer-based approaches face fundamental challenges that limit their effectiveness in time series forecasting. A central issue lies in the design of input tokenization strategies. As illustrated in Figure 1-(a), traditional methods employ point-wise tokenization (pixel-level), patch-wise tokenization (fixed-length sub-series), or series-wise tokenization (entire sequence as a single token). Point-wise tokenization, while fine-grained, incurs prohibitive computational costs due to the quadratic complexity of self-attention with respect to sequence length. Patch-wise tokenization (Nie et al., 2023) reduces computational burden but may impede the model's ability to capture long-range dependencies effectively, as it imposes locality constraints. Furthermore, modeling the multivariate interactions using such methods is not straightforward and poses significant challenges. Series-wise tokenization (Liu et al., 2024) captures holistic temporal patterns but struggles with modeling local context and becomes impractical for long sequences or large datasets due to high computational demands. Furthermore, the current fixed-length path-based and whole-series tokenization methods present significant challenges for time series foundation models, either due to computational overhead or feasibility issues (Goswami et al., 2024; Das et al., 2023).

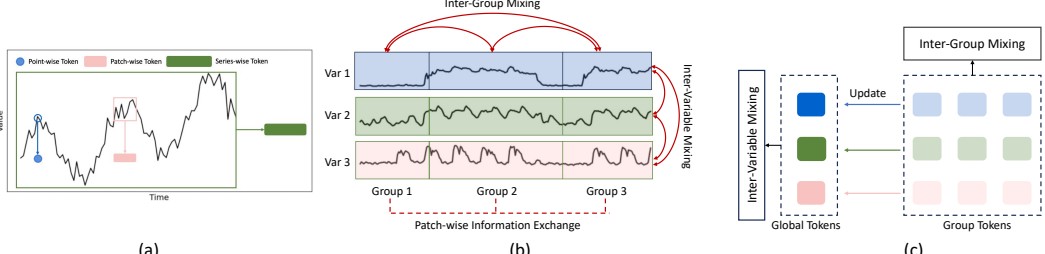

Figure 1: Overview of the TimeCAT framework. (a) Traditional tokenization approaches face challenges in balancing accuracy and computational cost. (b) Our context-aware approach dynamically groups sequences based on semantic content, enabling efficient hierarchical mixing at different levels of context. (c) The Token-Grouping-and-Merging module integrates global and group tokens to effectively capture both local and global temporal dependencies.

To address these challenges, we observe that time series data inherently exhibit hierarchical temporal structures and varying semantic characteristics across different segments. This insight motivates a context-aware approach where sequences are dynamically partitioned into semantically coherent groups based on the input context. By doing so, we can model both local patterns within groups and global trends across the entire sequence more effectively.

In this paper, we introduce **TimeCAT**—a Time series Context-Aware Transformer designed to capture complex temporal dependencies while maintaining computational efficiency. As depicted in Figure 1-(b), our approach dynamically groups sequences based on their semantic content, enabling efficient hierarchical mixing at different levels of context. We propose a novel *Context-Aware Mixing Block* that facilitates three hierarchical levels of information exchange: *(1) Intra-Group Mixing* focuses on capturing local dependencies within each group by applying self-attention mechanisms to tokens specific to that group. *(2) Inter-Group Mixing* enables interactions across different groups through mixing layers that aggregate information, thereby enhancing the model's ability to learn cross-group dependencies. Finally, *(3) Global-Level Mixing* incorporates a global token that collects information from all groups, capturing overarching temporal trends and facilitating interactions across variables throughout the entire sequence.

Our hierarchical processing scheme exploits both local and global patterns in a structured manner, addressing the limitations of traditional tokenization strategies. As illustrated in Figure 1-(c), the integration of group tokens and a global token allows TimeCAT to holistically model temporal dynamics, effectively balancing computational efficiency with accurate representation learning. By capturing rich interactions across different token types and levels of context, TimeCAT enhances the modeling of complex temporal patterns inherent in time series data. Our work makes the following key contributions:

- We propose a novel *dynamic grouping mechanism* that segments time series data into semantically meaningful groups based on input context, enabling efficient intra-group and inter-group interactions without compromising temporal context.

- We introduce a *Context-Aware Transformer* architecture that utilizes a hierarchical mixing block, facilitating fine-grained information exchange across intra-group, inter-group, and global levels. This design captures complex temporal patterns while significantly reducing computational complexity.

- Extensive experiments on challenging real-world datasets demonstrate that TimeCAT outperforms state-of-the-art models in both forecasting accuracy and computational efficiency, validating its effectiveness in modeling complex time series data.

By addressing the fundamental challenges in time series forecasting with Transformers, TimeCAT sets a new direction for efficient and accurate modeling of temporal data. Our approach leverages hierarchical context and dynamic grouping to overcome the limitations of existing methods, making it well-suited for real-world applications requiring long-horizon forecasting, including the time series foundation models.

## 2 RELATED WORK

**Transformer and MLP-based Time Series Forecasting**   Transformers have been widely adopted in time series forecasting (Zhou et al., 2021; 2022b; Zhang & Yan, 2023; Chen et al., 2021; 2024). PatchTST (Nie et al., 2022) introduced a Transformer architecture that splits input time series into fixed-length patches, applying self-attention for temporal information extraction. However, PatchTST lacks cross-channel interactions. Extensions have addressed these limitations, such as varied patch sizes for multi-resolution representations (Zhang et al., 2024), and representing the entire time series as a single token to capture holistic information (Liu et al., 2024). While the latter effectively models inter-variable interactions, it may lose temporal dynamics and is impractical for long sequences or large datasets. Alternatively, TimeMixer (Wang et al., 2024) uses a pure MLP-based mixing module to explore multi-scale representation learning, showing that distinct temporal patterns enhance forecasting. However, building large foundational time series models with MLP backbones poses challenges (Liang et al., 2024).

Recently, tokenization has gained attention as a crucial element in Transformer-based foundation models (Qian et al., 2022). MOMENT, a family of time series foundation models, emphasizes tokenization to model temporal dynamics (Goswami et al., 2024). Das et al. (2023) propose a decoder-only Transformer focusing on efficient token structures, while Garza & Mergenthaler-Canseco (2023) introduce TimeGPT-1, achieving state-of-the-art results by leveraging tokenization to model complex temporal relationships. These works highlight tokenization's key role in developing robust Transformer-based time series models, which still primarily focus on fixed-length tokenization.

**Token Merging & Clustering Methods**   To enhance token-based foundation models' efficiency, various token merging and clustering approaches have been proposed in both time series and image domains. Götz et al. (2024) introduce a token merging mechanism to reduce complexity by grouping tokens, significantly speeding up pretrained models on multivariate time series datasets but acting as a low-pass filter, potentially degrading prediction accuracy. In the image domain, "ToMe" (Bolya et al., 2022) merges redundant tokens in Vision Transformers (ViT) for a speed-accuracy trade-off without retraining. Additionally, Fan et al. (2024) propose clustering tokens based on semantic relevance, reducing computational costs but potentially affecting performance due to limited inter-cluster interactions and reliance on a global token.

These works inspire our approach, which integrates context-aware token grouping within a Transformer-based backbone to enable multi-scale information interactions. Unlike fine-grained, fixed-length patch-based methods (Nie et al., 2023) or coarse-grained whole-sequence tokenization (Liu et al., 2024), our approach introduces novel intra-group, inter-group, and global-level operations for efficient, fine-grained interactions.

## 3 TIMECAT

### 3.1 PROBLEM FORMULATION

Given a historical multivariate time series: $\mathbf{X} = [\mathbf{x}_1, \mathbf{x}_2, \ldots, \mathbf{x}_T]^\top \in \mathbb{R}^{T \times N}$, where $T$ is the number of time steps, $N$ is the number of variables, and $\mathbf{x}_t \in \mathbb{R}^N$ represents the observation at time $t$. Our goal is to predict future values over a forecast horizon $Q$: $\mathbf{Y} = [\mathbf{x}_{T+1}, \mathbf{x}_{T+2}, \ldots, \mathbf{x}_{T+Q}]^\top \in \mathbb{R}^{Q \times N}$.

### 3.2 MODEL ARCHITECTURE OVERVIEW

We propose TimeCAT, a novel Transformer-based architecture tailored for time series forecasting, illustrated in Figure 2. Building upon the encoder-only architecture of the Transformer (Vaswani, 2017), TimeCAT introduces key innovations to better capture temporal dependencies and reduce computational complexity.

Given the input sequence $\mathbf{X} \in \mathbb{R}^{T \times N}$, we first apply instance normalization to obtain the normalized sequence $\tilde{\mathbf{X}}$. The normalized time series is then divided into overlapping patches of length $L_p$ with the given stride , resulting in $P$ patches per variable. Each patch is embedded via a Multi-Layer Perceptron (MLP) (Nie et al., 2023), producing a sequence of embeddings per variable:

$$\mathbf{E}_n = [\mathbf{e}_n^{(1)}, \mathbf{e}_n^{(2)}, \ldots, \mathbf{e}_n^{(P)}]^\top \in \mathbb{R}^{P \times d},$$

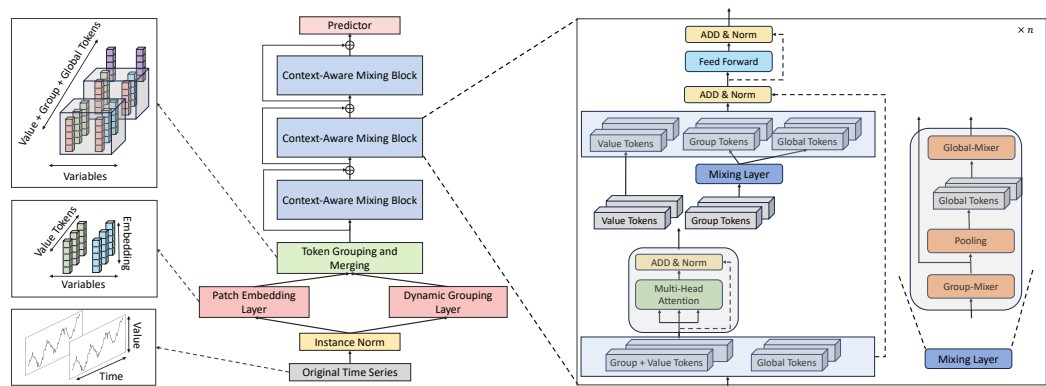

Figure 2: Architecture of the proposed TimeCAT framework. The original time series data undergoes instance normalization and is divided into value tokens through the Patch Embedding Layer. The Dynamic Grouping Layer further processes these tokens, enabling token grouping and merging. The resulting tokens—value, group, and global tokens—are processed by multiple Context-Aware Mixing Blocks, which facilitate interactions across different token types using multi-head attention and mixing layers. The right panel provides a detailed view of the Context-Aware Mixing Block, including the steps of intra-group, inter-group, and global information mixing. The Predictor module at the top aggregates the final mixed representations to produce the desired output.

where $d$ is the embedding dimension, and $n \in [1, N]$ indexes the variables.

To preserve temporal order, positional embeddings $\{\mathbf{p}_i\}_{i=1}^P$, where $\mathbf{p}_i \in \mathbb{R}^d$, are added:

$$\mathbf{E}_n = [\mathbf{e}_n^{(1)} + \mathbf{p}_1, \mathbf{e}_n^{(2)} + \mathbf{p}_2, \ldots, \mathbf{e}_n^{(P)} + \mathbf{p}_P]^\top.$$

The *Dynamic Grouping Layer* then partitions the sequence into context-aware groups and augments the embeddings with group and global tokens. The *Context-Aware Mixing Blocks* further process these tokens, enabling efficient intra-group, inter-group, and global interactions. Finally, the *Predictor Module* aggregates the representations to produce the final predictions, following the procedure in Nie et al. (2023).

## 3.3 DYNAMIC GROUPING AND TOKEN AUGMENTATION

**Dynamic Grouping**  To efficiently capture both local and global patterns, we dynamically partition the sequence into $G$ groups based on the input context. The grouping is determined by a learnable function that computes group ratios $\mathbf{r} \in \mathbb{R}^G$:

$$\mathbf{v} = \text{Flatten}(\tilde{\mathbf{X}}') \in \mathbb{R}^{(TN)/RD}, \quad \mathbf{r} = \text{Softmax}(\mathbf{W}_g \mathbf{v} + \mathbf{b}_g), \tag{1}$$

where $\tilde{\mathbf{X}}'$ is a downsampled version of $\tilde{\mathbf{X}}$ with downsampling ratio $RD$, $\mathbf{W}_g \in \mathbb{R}^{G \times (TN)/RD}$, and $\mathbf{b}_g \in \mathbb{R}^G$. The down-sampling is applied since its capability and efficiency to split the group.

The group sizes $\{s_i\}_{i=1}^G$ are computed as:

$$s_i = \lceil r_i \cdot P \rceil, \quad \text{subject to} \quad \sum_{i=1}^G s_i = P. \tag{2}$$

Group indices Indices$_i$ are then determined to segment the sequence.

**Token Augmentation**  To enrich the model's capacity to capture hierarchical contexts, we introduce a *Global Token* $\mathbf{g}_n \in \mathbb{R}^d$ and *Group Tokens* $\{\mathbf{g}_{n,i}\}_{i=1}^G$ for each variable $n$, where:

$$\mathbf{g}_{n,i} = \mathbf{g}'_{n,i} + \mathbf{l}_{s_i}, \quad \mathbf{g}'_{n,i} \in \mathbb{R}^d, \tag{3}$$

and $\mathbf{l}_{s_i}$ is a learnable embedding corresponding to the group size $s_i$.

The augmented sequence for variable $n$ becomes:

$$\mathbf{S}_n = [\mathbf{g}_n; \mathbf{g}_{n,1}; \ldots; \mathbf{g}_{n,G}; \mathbf{E}_n] \in \mathbb{R}^{(1+G+P)\times d}. \tag{4}$$

The overall sequence for all variables is:

$$\mathbf{S} = [\mathbf{S}_1; \mathbf{S}_2; \ldots; \mathbf{S}_N] \in \mathbb{R}^{N\times(1+G+P)\times d}. \tag{5}$$

This transformation from the initial input sequence to the representation after patch embedding, and eventually to our employed combined representation, is illustrated in the left part of Figure 2.

### 3.4 CONTEXT-AWARE MIXING BLOCK

The *Context-Aware Mixing Block* is designed to efficiently model both local and global dependencies by processing the sequence in a hierarchical manner. This block enhances sequence modeling by capturing rich interactions across different token types, enabling improved learning of relationships at varying granularities.

**Input Partitioning**  For each variable $n$, the augmented sequence $\mathbf{S}_n$ is partitioned into:

- **Global Token**: $x_{\text{global},n} \in \mathbb{R}^d$,
- **Group Tokens**: $x_{\text{group},n} = [\mathbf{g}_{n,1}, \ldots, \mathbf{g}_{n,G}]^\top \in \mathbb{R}^{G\times d}$,
- **Value Tokens**: $x_{\text{value},n} = [\mathbf{e}_n^{(1)}, \ldots, \mathbf{e}_n^{(P)}]^\top \in \mathbb{R}^{P\times d}$.

We further partition the value tokens into groups based on the indices from equation 2, resulting in $G$ groups $\{x_{\text{value},n}^i\}_{i=1}^G$, where $x_{\text{value},n}^i \in \mathbb{R}^{s_i\times d}$.

**Intra-Group Operations**  Within each group $i$, we concatenate the corresponding group token $\mathbf{g}_{n,i}$ with the value tokens $x_{\text{value},n}^i$:

$$x_{\text{concat},n}^i = [\mathbf{g}_{n,i}; x_{\text{value},n}^i] \in \mathbb{R}^{(1+s_i)\times d}.$$

Self-attention is then applied to model local dependencies:

$$\tilde{x}_n^i = \text{SelfAttention}(x_{\text{concat},n}^i). \tag{6}$$

Residual connections and layer normalization are applied:

$$\hat{\mathbf{g}}_{n,i} = \text{LayerNorm}(\mathbf{g}_{n,i} + \tilde{\mathbf{g}}_{n,i}), \tag{7}$$

$$\hat{x}_{\text{value},n}^i = \text{LayerNorm}(x_{\text{value},n}^i + \tilde{x}_{\text{value},n}^i). \tag{8}$$

This process refines the group and value tokens by capturing local context within each group, enhancing the model's ability to learn fine-grained patterns.

**Inter-Group and Global Operations**  To enable effective communication and capture dependencies across different groups, we process the group tokens $\{\hat{\mathbf{g}}_{n,i}\}_{i=1}^G$:

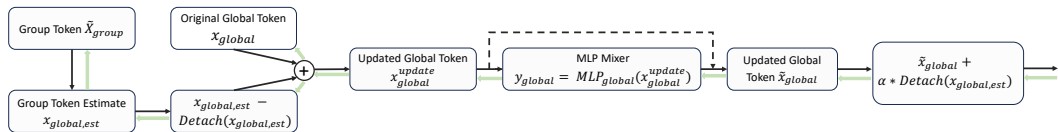

Figure 3: The Global Token update mechanism. The black lines represent the forward information flow, while the green lines indicate the backward gradient flow. Gradient detachment prevents the global token from dominating the learning process, ensuring balanced representation learning.

1. **Group Mixing**: We transpose and apply an MLP mixer to allow cross-group interactions:

$$Y_{\text{group},n} = \text{MLP}_{\text{group}}(\hat{X}_{\text{group},n}^{\top}), \quad \hat{X}_{\text{group},n}^{\top} \in \mathbb{R}^{d \times G}, \tag{9}$$

where $\hat{X}_{\text{group},n} = [\hat{\mathbf{g}}_{n,1}, \ldots, \hat{\mathbf{g}}_{n,G}]^{\top}$.

The output is transposed back and residual connections with layer normalization are applied:

$$\tilde{X}_{\text{group},n} = \text{LayerNorm}(\hat{X}_{\text{group},n} + Y_{\text{group},n}^{\top}). \tag{10}$$

This operation facilitates interactions between different groups, allowing the model to capture higher-level patterns and dependencies.

2. **Global Token Update**: We first aggregate the refined group tokens to estimate a global representation:

$$x_{\text{global, est},n} = \text{Pooling}(\tilde{X}_{\text{group},n}). \tag{11}$$

A naive update of the global token using $x_{\text{global, est},n}$ may lead to training instability due to large gradients. To address this, we apply gradient detachment:

$$x_{\text{global},n}^{\text{updated}} = x_{\text{global},n} + \left(x_{\text{global, est},n} - \text{Detach}(x_{\text{global, est},n})\right). \tag{12}$$

An MLP mixer and residual connections with layer normalization are applied:

$$y_{\text{global},n} = \text{MLP}_{\text{global}}(x_{\text{global},n}^{\text{updated}}), \tag{13}$$

$$\tilde{x}_{\text{global},n} = \text{LayerNorm}(x_{\text{global},n}^{\text{updated}} + y_{\text{global},n}). \tag{14}$$

Finally, we combine the global token with a detached version of the global estimate:

$$\tilde{x}_{\text{global},n} = \tilde{x}_{\text{global},n} + \alpha \cdot \text{Detach}(x_{\text{global, est},n}), \tag{15}$$

where $\alpha$ is a learnable parameter. This mechanism balances the influence of global and local contexts while ensuring stable training dynamics.

The global token update mechanism is illustrated in Figure 3. By controlling the gradient flow, we prevent the global token from overwhelming the learning of local details, allowing the model to converge faster and more stably while effectively capturing both global context and nuanced local information. The effectiveness of our approach is evaluated in the ablation study.

**Sequence Reconstruction and Post-processing** The refined tokens are concatenated to reconstruct the sequence:

$$X_n = [\tilde{x}_{\text{global},n}; \tilde{X}_{\text{group},n}; \hat{X}_{\text{value},n}] \in \mathbb{R}^{(1+G+P) \times d}. \tag{16}$$

Residual connections and layer normalization are applied for stability. A feed-forward network (FFN) with residual connections introduces non-linearity and further refines the representation. This reconstructed sequence is then used in subsequent Context-Aware Mixing Blocks or passed to the Predictor module.

Table 1: Long-term forecasting results. All the results are averaged from 4 different prediction lengths, that is $\{96, 192, 336, 720\}$. A lower MSE or MAE indicates a better prediction. We fix the input length as 96 for all experiments. See Table 5 in Appendix for the full results.

| Models | TimeCAT (Ours) | | iTransformer ((2024)) | | TimeMixer (2024) | | PatchTST (2023) | | TimesNet (2023) | | Crossformer (2023) | | MICN (2023) | | FiLM (2022a) | | DLinear (2023) | | FEDformer (2022b) | |
|---|---|---|---|---|---|---|---|---|---|---|---|---|---|---|---|---|---|---|---|---|
| Metric | MSE | MAE | MSE | MAE | MSE | MAE | MSE | MAE | MSE | MAE | MSE | MAE | MSE | MAE | MSE | MAE | MSE | MAE | MSE | MAE |
| Weather | **0.238** | **0.267** | 0.258 | 0.278 | 0.240 | 0.271 | 0.265 | 0.285 | 0.251 | 0.294 | 0.264 | 0.320 | 0.268 | 0.321 | 0.271 | 0.291 | 0.265 | 0.315 | 0.309 | 0.360 |
| Electricity | **0.172** | **0.265** | 0.182 | 0.270 | 0.178 | 0.272 | 0.216 | 0.318 | 0.193 | 0.304 | 0.244 | 0.334 | 0.196 | 0.309 | 0.223 | 0.302 | 0.225 | 0.319 | 0.214 | 0.327 |
| Traffic | **0.408** | **0.271** | 0.428 | 0.282 | 0.484 | 0.297 | 0.529 | 0.341 | 0.620 | 0.336 | 0.667 | 0.426 | 0.593 | 0.356 | 0.637 | 0.384 | 0.625 | 0.383 | 0.610 | 0.376 |
| ETTh1 | **0.422** | **0.430** | 0.447 | 0.447 | 0.454 | 0.447 | 0.516 | 0.484 | 0.495 | 0.450 | 0.529 | 0.522 | 0.475 | 0.480 | 0.516 | 0.483 | 0.461 | 0.457 | 0.498 | 0.484 |
| ETTh2 | **0.364** | **0.394** | 0.364 | 0.395 | 0.383 | 0.407 | 0.391 | 0.411 | 0.414 | 0.427 | 0.942 | 0.684 | 0.574 | 0.531 | 0.402 | 0.420 | 0.563 | 0.519 | 0.437 | 0.449 |
| ETTm1 | **0.377** | **0.392** | 0.381 | 0.395 | 0.407 | 0.410 | 0.406 | 0.407 | 0.400 | 0.406 | 0.513 | 0.495 | 0.423 | 0.422 | 0.411 | 0.402 | 0.404 | 0.408 | 0.448 | 0.452 |
| ETTm2 | **0.272** | **0.318** | 0.275 | 0.323 | 0.288 | 0.332 | 0.290 | 0.334 | 0.291 | 0.333 | 0.757 | 0.610 | 0.353 | 0.402 | 0.287 | 0.329 | 0.354 | 0.402 | 0.305 | 0.349 |

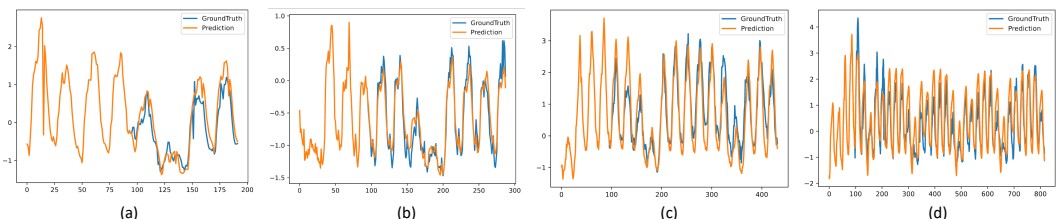

(a)  (b)  (c)  (d)

Figure 4: Visualization of ECL prediction using TimeCAT for next (a) 96; (b) 192; (c) 336 and (d) 720 steps.

**Computational Complexity Reduction** By applying self-attention within groups rather than across the entire sequence, we significantly reduce computational complexity. The relative reduction $\mathcal{R}$ in complexity is:

$$\mathcal{R} = 1 - \frac{\mathcal{O}_{\text{group}}}{\mathcal{O}_{\text{full}}} = 1 - \left( \frac{1}{G} + \frac{2}{P} + \frac{G}{P^2} \right), \tag{17}$$

where $\mathcal{O}_{\text{group}}$ and $\mathcal{O}_{\text{full}}$ denote the complexities with grouping and without grouping, respectively. In scenarios where $P \gg G$, increasing $G$ leads to higher computational savings, as demonstrated in our experiments. This efficiency gain allows TimeCAT to handle longer sequences without incurring prohibitive computational costs. Details are in Appendix A.3.1.

## 4 EXPERIMENTS

We extensively include 7 real-world datasets in our experiments, including ETT (4 subsets), Electricity, Traffic, Weather used by Autoformer (Chen et al., 2021). Detailed dataset descriptions are provided in Appendix A.1.

**Baselines** We carefully choose 9 well-acknowledged forecasting models as our benchmark, including (1) Transformer-based methods: iTransformer (Liu et al., 2024), Autoformer (Chen et al., 2021), FEDformer (Zhou et al., 2022b), Stationary (Liu et al., 2022), Crossformer (Zhang & Yan, 2023), PatchTST (Nie et al., 2023); (2) Linear-based methods: DLinear (Zeng et al., 2023); (3) TCN-based method TimesNet (Wu et al., 2023) and (4) MLP-Mixer based method (Wang et al., 2024).

### 4.1 MAIN RESULTS

**Long-term Forecasting Results.** Table 1 compares the forecasting performance of various models across multiple datasets. TimeCAT consistently achieves the best or second-best results in both MSE

Table 2: Ablation study. **W/O Adap**, **W/O Group-Mix**, **W/O Global-Mix**, **W/O Skip-Connect**, and **W/O** $E_{len}$ represent removing the adaptive grouping mechanism, group mixing layer, global mixing layer, skip connections, and embedding length parameter, respectively. The final column shows the performance of the full TimeCAT model. Lower MSE and MAE values indicate better forecasting performance.

| Models | | W/O Adap | | W/O Group-Mix | | W/O Global-Mix | | W/O Skip-Connect | | W/O $E_{len}$ | | TimeCAT | |
|---|---|---|---|---|---|---|---|---|---|---|---|---|---|
| Metric | | MSE | MAE | MSE | MAE | MSE | MAE | MSE | MAE | MSE | MAE | MSE | MAE |
| Weather | 96 | 0.160 | 0.205 | 0.165 | 0.210 | 0.158 | 0.203 | 0.162 | 0.207 | 0.164 | 0.209 | **0.153** | **0.199** |
| | 192 | 0.215 | 0.252 | 0.220 | 0.258 | 0.208 | 0.245 | 0.213 | 0.250 | 0.210 | 0.247 | **0.204** | **0.247** |
| | 336 | 0.270 | 0.295 | 0.275 | 0.300 | 0.265 | 0.290 | 0.268 | 0.292 | 0.262 | 0.290 | **0.261** | **0.289** |
| | 720 | 0.345 | 0.350 | 0.355 | 0.360 | 0.340 | 0.345 | 0.342 | 0.347 | 0.340 | 0.344 | **0.337** | **0.336** |
| Electricity | 96 | 0.155 | 0.250 | 0.160 | 0.255 | 0.150 | 0.245 | 0.152 | 0.248 | 0.154 | 0.250 | **0.148** | **0.245** |
| | 192 | 0.170 | 0.260 | 0.175 | 0.265 | 0.165 | 0.255 | 0.168 | 0.258 | 0.162 | 0.253 | **0.163** | **0.253** |
| | 336 | 0.185 | 0.280 | 0.190 | 0.285 | 0.180 | 0.275 | 0.182 | 0.277 | 0.178 | 0.272 | **0.176** | **0.271** |
| | 720 | 0.210 | 0.315 | 0.220 | 0.325 | 0.205 | 0.310 | 0.208 | 0.315 | 0.203 | 0.308 | **0.200** | **0.292** |

and MAE, outperforming other transformer-based models such as iTransformer and PatchTST across most datasets.

**TimeCAT vs. iTransformer:** TimeCAT demonstrates strong performance improvements over iTransformer across all datasets, with an average reduction of 7.8% in MSE and 4.0% in MAE. For instance, on **Weather**, TimeCAT achieves lower MSE (0.238 vs. 0.258) and MAE (0.267 vs. 0.278). On **Electricity**, TimeCAT shows consistent reductions in error (0.172 MSE vs. 0.182, 0.265 MAE vs. 0.270). These results indicate that TimeCAT's context-aware mechanisms significantly enhance predictive accuracy over the standard transformer framework of iTransformer.

**TimeCAT vs. PatchTST:** While PatchTST improves upon iTransformer with patch-based tokenization, TimeCAT achieves an average reduction of 5.4% in MSE and 4.6% in MAE compared to PatchTST. For example, on **Weather**, TimeCAT achieves significantly lower MSE (0.238 vs. 0.265) and MAE (0.267 vs. 0.285). Across other datasets like **ETTh1** and **ETTm2**, TimeCAT consistently outperforms PatchTST, reinforcing its robustness in handling temporal dependencies.

**General Comparison:** TimeCAT also outperforms non-transformer models such as TimeMixer, TimesNet, and Crossformer across all datasets. On challenging datasets like **Traffic**, Time-CAT achieves substantial reductions in both MSE (0.408 vs. Crossformer's 0.667) and MAE (0.271 vs. 0.426). This highlights TimeCAT's strong capability in multivariate time series forecasting.

Figure 4 illustrates the effectiveness of the TimeCAT model in predicting electricity consumption levels (ECL) across various forecasting horizons. The visualization demonstrates the model's predictions alongside the actual ground truth data for next 96, 192, 336, and 720 steps, respectively. Each graph shows that TimeCAT adeptly captures the trends and fluctuations of the data, maintaining high accuracy and consistency in both shorter and longer-term forecasts. The model's performance is especially notable in the longest forecast of 720 steps, where it continues to closely align with the ground truth, showcasing its robustness and reliability in multi-step time series forecasting. Full comparision with iTransformer (Liu et al., 2024) and PatchTST (Nie et al., 2023) is in Appendix A.4.

**Overall Summary:** TimeCAT shows substantial reductions in forecasting errors compared to state-of-the-art models. The context-aware transformer design and dynamic grouping strategies allow it to better capture temporal dependencies, achieving an average reduction of 6.6% in MSE and 4.3% in MAE over both iTransformer and PatchTST.

## 4.2 MODEL ANALYSIS

**Ablation Study** To evaluate the contribution of each module in the TimeCAT framework, we conducted an ablation study by systematically removing key components and assessing their impact on forecasting performance. Specifically, we examined the effects of eliminating the adaptive grouping mechanism (**W/O Adap**), the group mixing layer (**W/O Group-Mix**), the global mixing layer (**W/O Global-Mix**), the skip connections (**W/O Skip-Connect**), and the embedding length

Table 3: Parameter sensitivity study. The prediction accuracy varies with the number of groups $G$. Lower MSE and MAE values indicate better forecasting performance.

| Metric | | $G = 2$ | | $G = 3$ | | $G = 4$ | | $G = 5$ | |
|---|---|---|---|---|---|---|---|---|---|
| | | MSE | MAE | MSE | MAE | MSE | MAE | MSE | MAE |
| Weather | 96 | **0.153** | **0.199** | 0.154 | 0.200 | 0.155 | 0.201 | 0.156 | 0.202 |
| | 192 | **0.204** | **0.247** | 0.205 | 0.248 | 0.206 | 0.249 | 0.207 | 0.250 |
| | 336 | **0.261** | **0.289** | 0.262 | 0.290 | 0.263 | 0.291 | 0.264 | 0.292 |
| | 720 | **0.337** | **0.336** | 0.338 | 0.337 | 0.339 | 0.338 | 0.340 | 0.339 |
| Electricity | 96 | 0.149 | 0.244 | **0.148** | **0.245** | 0.149 | 0.247 | 0.151 | 0.248 |
| | 192 | 0.164 | 0.254 | **0.163** | **0.253** | 0.164 | 0.255 | 0.166 | 0.256 |
| | 336 | 0.177 | 0.271 | **0.176** | **0.271** | 0.178 | 0.273 | 0.179 | 0.274 |
| | 720 | 0.201 | 0.293 | **0.200** | **0.292** | 0.202 | 0.294 | 0.203 | 0.295 |

parameter ($E_{len}$) (**W/O** $E_{len}$). Table 2 presents the MSE and MAE results across different forecasting horizons for the Weather and Electricity datasets.

The results indicate that each component plays a critical role in enhancing the model's forecasting accuracy. Removing the adaptive grouping mechanism leads to a noticeable degradation in performance, highlighting its importance in dynamically partitioning the time series into semantically coherent groups. Similarly, omitting the group mixing and global mixing layers results in increased errors, underscoring their roles in facilitating intra-group and inter-group interactions as well as capturing global temporal dependencies. The absence of skip connections and the embedding length parameter also adversely affects the model's performance, albeit to a lesser extent. These findings collectively demonstrate that the integrated modules of TimeCAT synergistically contribute to its superior forecasting capabilities.

**Parameter Sensitivity Study**  To investigate the impact of the number of groups $G$ on the forecasting performance of TimeCAT, we conducted a parameter sensitivity study. We evaluated the model across different values of $G$ (i.e., 2, 3, 4, and 5) on two real-world datasets: Weather and Electricity. Table 3 presents the Mean Squared Error (MSE) and Mean Absolute Error (MAE) results for various forecasting horizons.

The results indicate a clear dependency of the model's performance on the choice of $G$. For the Weather dataset, setting $G = 2$ achieves the lowest MSE and MAE across all forecasting horizons, suggesting that two groups are sufficient to capture the underlying temporal patterns without introducing unnecessary complexity. Increasing $G$ beyond 2 leads to a slight decline in performance, likely due to over-segmentation and the introduction of minor noise.

Conversely, for the Electricity dataset, $G = 3$ consistently yields the best performance across all forecasting horizons. This optimal group number effectively balances the trade-off between capturing intricate temporal dependencies and maintaining computational efficiency. Selecting $G$ values lower or higher than 3 results in marginally increased forecasting errors, indicating that three groups best represent the semantic structures inherent in the Electricity data.

These findings underscore the importance of appropriately selecting the number of groups $G$ to align with the dataset's characteristics, thereby enhancing the model's forecasting accuracy and efficiency.

**Analysis of Learned Group and Global Tokens**  Figure 5 highlights the effectiveness of our grouping strategy in the ETTh1 dataset. Figure 5-(a) shows the input data's correlation matrix, revealing inherent dependencies among variables, while Figure 5-(b) illustrates the correlation matrix of learned global tokens, closely mirroring the input. This alignment confirms that the global tokens capture essential temporal patterns, validating our context-aware approach.

The t-SNE plot in Figure 5-(c) reveals distinct clusters of global tokens, reflecting effective separation of variables and high-level interactions. Figures 5-(d) and 5-(e) display the first and second groups of tokens, showing tightly clustered variables consistent with the correlations in Figure 5-(a).

Our skip-connect global token learning mechanism integrates global and group tokens, enhancing the model's capacity to capture complex dependencies and improving forecasting accuracy. These

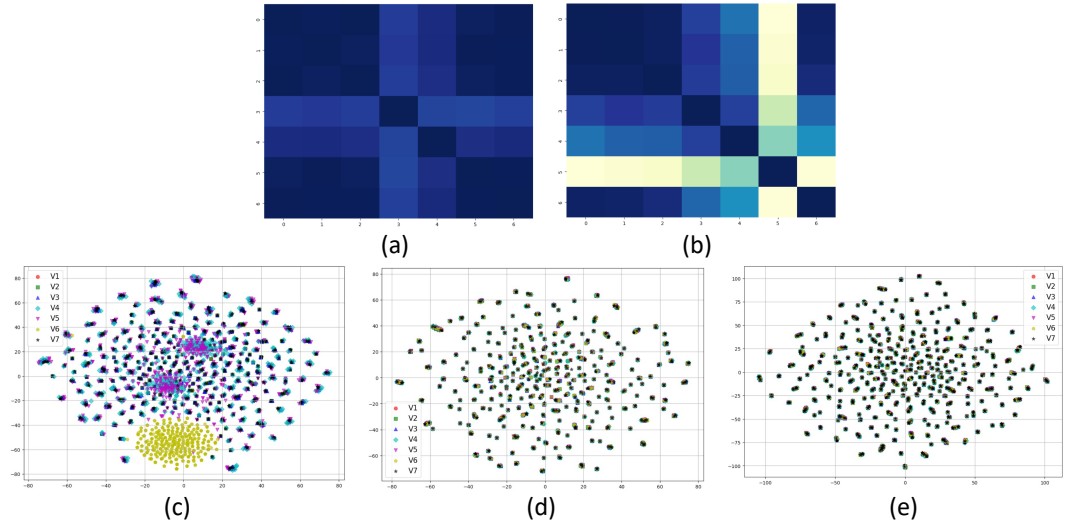

Figure 5: Visualization of learned global and group tokens for the ETTh1 dataset. (a) Correlation matrix of the input data; (b) Correlation matrix of the global tokens; (c) Visualization of global tokens colored by variable categories; (d) Visualization of the first group of tokens; (e) Visualization of the second group of tokens.

visualizations confirm that our grouping mechanism and skip connections effectively model both global summaries and local patterns within the time series.

## 5 CONCLUSION

We introduced **TimeCAT** (Time series Context-Aware Transformer), a Transformer-based model enhancing time series forecasting through dynamic grouping and hierarchical mixing. Our ablation studies confirmed that these features significantly improve forecasting accuracy and efficiency. Optimal group numbers, determined through parameter sensitivity analysis, consistently boosted performance across datasets. Visualizations of global and group tokens validated that TimeCAT effectively organizes variables into clusters and captures intricate relationships. Comparative experiments showed TimeCAT outperforms leading models, setting a new standard in the field. Future efforts will refine adaptive grouping mechanisms, extend handling capabilities for diverse data, and explore scalability for large-scale applications. Overall, TimeCAT advances efficient, accurate forecasting, promising further innovations in temporal modeling.

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

## A APPENDIX

### A.1 IMPLEMENTATION DETAILS

**Dataset**  Our research utilizes seven diverse, real-world datasets to evaluate the efficacy of our newly introduced model, TimeCAT. These datasets include the ETT dataset, which features data from electricity transformers, covering seven distinct variables from July 2016 to July 2018, split into four segments: ETTh1 and ETTh2 with hourly intervals, and ETTm1 and ETTm2 with 15-minute intervals. The Weather dataset is derived from the Max Planck Institute for Biogeochemistry's weather station in 2020, comprising 21 weather parameters measured at 10-minute intervals. The ECL dataset consists of electricity usage data for 321 clients on an hourly basis. Additionally, the Traffic dataset encompasses data from 862 sensors, monitoring hourly traffic occupancy rates on freeways in the San Francisco Bay area, spanning from January 2015 to December 2016. Our experimental methodology strictly adheres to the data preprocessing and splitting protocols established by iTransformer Liu et al. (2024) to prevent any data leakage, with the datasets segmented chronologically into training, validation, and test sets. We apply forecasting models that utilize a historical lookback window of 96 time points, and we test prediction intervals of {96, 192, 336, 720}. Below, we provide a comprehensive table outlining the specific attributes of each dataset:

Table 4: Comprehensive dataset attributes. *Variate Count* indicates the number of variables in each dataset. *Total Data Points* provides the number of time points in each phase of the (Train, Validation, Test) split. *Forecast Horizon* lists the different prediction durations. *Proportion* shows the division ratio for training, validation, and test sets. *Interval* denotes the time between each data point.

| Dataset | Variate Count | Forecast Horizon | Total Data Points | Proportion | Interval | Sector |
|---|---|---|---|---|---|---|
| ETTh1, ETTh2 | 7 | {96, 192, 336, 720} | (8545, 2881, 2881) | 60:20:20 | Hourly | Electricity |
| ETTm1, ETTm2 | 7 | {96, 192, 336, 720} | (34465, 11521, 11521) | 60:20:20 | Every 15 min | Electricity |
| Weather | 21 | {96, 192, 336, 720} | (36792, 5271, 10540) | 70:10:20 | Every 10 min | Weather |
| ECL | 321 | {96, 192, 336, 720} | (18317, 2633, 5261) | 70:10:20 | Hourly | Electricity |
| Traffic | 862 | {96, 192, 336, 720} | (12185, 1757, 3509) | 70:10:20 | Hourly | Transportation |

**Implementation Details**  All experiments were run three times, implemented in Pytorch, and conducted on a single NVIDIA A100 80GB GPU. Most of the compared baseline models that we reproduced are implemented based on the benchmark of TimesNet (Wu et al., 2023), while iTransformer Liu et al. (2024) and TimeMixer (Wang et al., 2024) is based on their public repository and settings. We set the initial learning rate as $10^{-2}$ or $10^{-3}$ and used the ADAM optimizer (Kingma, 2014) with L2 loss for model optimization. And the batch size was set to be 8 between 128. And we also provide the pseudo-code of TimeCAT in Algorithm 1. The source code will be open sourced and provided during the discussion period.

### A.2 FULL RESULTS

To ensure a fair comparison between models, we conducted experiments using unified parameters and reported results in the main text, including aligning all the input lengths, batch sizes, and training epochs in all experiments. Here, we provide the full results for each forecasting setting in Table 5.

### A.3 DISCUSSIONS

#### A.3.1 COMPUTATION EFFICIENCY

To analyze the computational reduction rate of self-attention modules due to the grouping mechanism, we assume, without loss of generality, that the groups are split uniformly.

The computational complexity of the self-attention mechanism without grouping is:

$$\mathcal{O}_{\text{no\_group}} = P^2 \cdot d$$

where $P$ is the number of patches, and $d$ is the embedding dimension.

With our grouping method, the number of patches per group is $s = P/G$, where $G$ is the number of groups. Additionally, one group token is added per group for self-attention information exchange, as

Table 5: Unified hyperparameter results for the long-term forecasting task. We compare extensive competitive models under different prediction lengths. *Avg* is averaged from all four prediction lengths, that is 96, 192, 336, 720.

| Models | | TimeCAT (Ours) | | TimeMixer 2024 | | iTransformer 2024 | | PatchTST 2023 | | TimesNet 2023 | | Crossformer 2023 | | MICN 2023 | | FiLM 2022a | | DLinear 2023 | | FEDformer 2022b | |
|---|---|---|---|---|---|---|---|---|---|---|---|---|---|---|---|---|---|---|---|---|---|
| Metric | | MSE | MAE | MSE | MAE | MSE | MAE | MSE | MAE | MSE | MAE | MSE | MAE | MSE | MAE | MSE | MAE | MSE | MAE | MSE | MAE |
| Weather | 96 | **0.153** | **0.199** | 0.163 | 0.209 | 0.174 | 0.214 | 0.186 | 0.227 | 0.172 | 0.220 | 0.195 | 0.271 | 0.198 | 0.261 | 0.195 | 0.236 | 0.195 | 0.252 | 0.217 | 0.296 |
| | 192 | **0.204** | **0.247** | 0.208 | 0.250 | 0.221 | 0.254 | 0.234 | 0.265 | 0.219 | 0.261 | 0.209 | 0.277 | 0.239 | 0.299 | 0.239 | 0.271 | 0.237 | 0.295 | 0.276 | 0.336 |
| | 336 | 0.261 | 0.289 | 0.251 | 0.287 | 0.278 | 0.296 | 0.284 | 0.301 | 0.246 | 0.337 | 0.273 | 0.332 | 0.285 | 0.336 | 0.289 | 0.306 | 0.282 | 0.331 | 0.339 | 0.380 |
| | 720 | **0.337** | **0.336** | 0.339 | 0.341 | 0.358 | 0.347 | 0.356 | 0.349 | 0.365 | 0.359 | 0.379 | 0.401 | 0.351 | 0.388 | 0.361 | 0.351 | 0.345 | 0.382 | 0.403 | 0.428 |
| | Avg | **0.238** | **0.267** | 0.240 | 0.271 | 0.258 | 0.278 | 0.265 | 0.285 | 0.251 | 0.294 | 0.264 | 0.320 | 0.268 | 0.321 | 0.271 | 0.291 | 0.265 | 0.315 | 0.309 | 0.360 |
| Electricity | 96 | **0.148** | 0.245 | 0.153 | 0.247 | 0.148 | **0.240** | 0.190 | 0.296 | 0.168 | 0.272 | 0.219 | 0.314 | 0.180 | 0.293 | 0.198 | 0.274 | 0.210 | 0.302 | 0.193 | 0.308 |
| | 192 | 0.163 | 0.253 | 0.166 | 0.256 | 0.162 | 0.253 | 0.199 | 0.304 | 0.184 | 0.322 | 0.231 | 0.322 | 0.189 | 0.302 | 0.198 | 0.278 | 0.210 | 0.305 | 0.201 | 0.315 |
| | 336 | **0.176** | 0.271 | 0.185 | 0.277 | 0.178 | 0.269 | 0.217 | 0.319 | 0.198 | 0.300 | 0.246 | 0.337 | 0.198 | 0.312 | 0.217 | 0.300 | 0.223 | 0.319 | 0.214 | 0.329 |
| | 720 | **0.200** | **0.292** | 0.225 | 0.310 | 0.225 | 0.317 | 0.258 | 0.352 | 0.220 | 0.320 | 0.280 | 0.363 | 0.217 | 0.330 | 0.278 | 0.356 | 0.258 | 0.350 | 0.246 | 0.355 |
| | Avg | **0.172** | **0.265** | 0.182 | 0.272 | 0.178 | 0.270 | 0.216 | 0.318 | 0.193 | 0.304 | 0.244 | 0.334 | 0.196 | 0.309 | 0.223 | 0.302 | 0.225 | 0.319 | 0.214 | 0.327 |
| Traffic | 96 | **0.381** | **0.257** | 0.462 | 0.285 | 0.395 | 0.268 | 0.526 | 0.347 | 0.593 | 0.321 | 0.644 | 0.429 | 0.577 | 0.350 | 0.647 | 0.384 | 0.650 | 0.396 | 0.587 | 0.366 |
| | 192 | **0.398** | **0.261** | 0.473 | 0.296 | 0.417 | 0.276 | 0.522 | 0.332 | 0.617 | 0.336 | 0.665 | 0.431 | 0.589 | 0.356 | 0.600 | 0.361 | 0.598 | 0.370 | 0.604 | 0.373 |
| | 336 | **0.418** | **0.275** | 0.498 | 0.296 | 0.433 | 0.283 | 0.517 | 0.334 | 0.629 | 0.336 | 0.674 | 0.420 | 0.594 | 0.358 | 0.610 | 0.367 | 0.605 | 0.373 | 0.621 | 0.383 |
| | 720 | **0.433** | **0.291** | 0.506 | 0.313 | 0.467 | 0.302 | 0.552 | 0.352 | 0.640 | 0.350 | 0.683 | 0.424 | 0.613 | 0.361 | 0.691 | 0.425 | 0.645 | 0.394 | 0.626 | 0.382 |
| | Avg | **0.408** | **0.271** | 0.484 | 0.297 | 0.428 | 0.282 | 0.529 | 0.341 | 0.620 | 0.336 | 0.667 | 0.426 | 0.593 | 0.356 | 0.637 | 0.384 | 0.625 | 0.383 | 0.610 | 0.376 |
| ETTh1 | 96 | 0.377 | **0.399** | 0.375 | 0.400 | 0.386 | 0.405 | 0.460 | 0.447 | 0.384 | 0.402 | 0.423 | 0.448 | 0.426 | 0.446 | 0.438 | 0.433 | 0.397 | 0.412 | 0.395 | 0.424 |
| | 192 | **0.418** | 0.425 | 0.429 | **0.421** | 0.441 | 0.436 | 0.512 | 0.477 | 0.436 | 0.429 | 0.471 | 0.474 | 0.454 | 0.464 | 0.493 | 0.466 | 0.446 | 0.441 | 0.469 | 0.470 |
| | 336 | **0.447** | **0.441** | 0.484 | 0.458 | 0.487 | 0.458 | 0.546 | 0.496 | 0.638 | 0.469 | 0.570 | 0.546 | 0.493 | 0.487 | 0.547 | 0.495 | 0.489 | 0.467 | 0.530 | 0.499 |
| | 720 | **0.446** | **0.458** | 0.498 | 0.482 | 0.503 | 0.491 | 0.544 | 0.517 | 0.521 | 0.500 | 0.653 | 0.621 | 0.526 | 0.526 | 0.586 | 0.538 | 0.513 | 0.510 | 0.598 | 0.544 |
| | Avg | **0.422** | **0.430** | 0.447 | 0.440 | 0.454 | 0.447 | 0.516 | 0.484 | 0.495 | 0.450 | 0.529 | 0.522 | 0.475 | 0.480 | 0.516 | 0.483 | 0.461 | 0.457 | 0.498 | 0.484 |
| ETTh2 | 96 | **0.287** | **0.339** | 0.289 | 0.341 | 0.297 | 0.349 | 0.308 | 0.355 | 0.340 | 0.374 | 0.745 | 0.584 | 0.372 | 0.424 | 0.322 | 0.364 | 0.340 | 0.394 | 0.358 | 0.397 |
| | 192 | **0.368** | **0.390** | 0.372 | 0.392 | 0.380 | 0.400 | 0.393 | 0.405 | 0.402 | 0.414 | 0.877 | 0.656 | 0.492 | 0.492 | 0.404 | 0.414 | 0.482 | 0.479 | 0.429 | 0.439 |
| | 336 | 0.390 | **0.411** | 0.386 | 0.414 | 0.428 | 0.432 | 0.427 | 0.436 | 0.452 | 0.452 | 1.043 | 0.731 | 0.607 | 0.555 | 0.435 | 0.445 | 0.591 | 0.541 | 0.496 | 0.487 |
| | 720 | **0.411** | 0.436 | 0.412 | **0.434** | 0.427 | 0.445 | 0.436 | 0.450 | 0.462 | 0.468 | 1.104 | 0.763 | 0.824 | 0.655 | 0.447 | 0.458 | 0.839 | 0.661 | 0.463 | 0.474 |
| | Avg | **0.364** | **0.394** | 0.364 | 0.395 | 0.383 | 0.407 | 0.391 | 0.411 | 0.414 | 0.427 | 0.942 | 0.684 | 0.574 | 0.531 | 0.402 | 0.420 | 0.563 | 0.519 | 0.437 | 0.449 |
| ETTm1 | 96 | **0.318** | **0.355** | 0.320 | 0.357 | 0.334 | 0.368 | 0.352 | 0.374 | 0.338 | 0.375 | 0.404 | 0.426 | 0.365 | 0.387 | 0.353 | 0.370 | 0.346 | 0.374 | 0.379 | 0.419 |
| | 192 | **0.358** | **0.375** | 0.361 | 0.381 | 0.377 | 0.391 | 0.390 | 0.393 | 0.374 | 0.387 | 0.450 | 0.451 | 0.403 | 0.408 | 0.389 | 0.387 | 0.382 | 0.391 | 0.426 | 0.441 |
| | 336 | **0.387** | **0.401** | 0.390 | 0.404 | 0.426 | 0.420 | 0.421 | 0.414 | 0.410 | 0.411 | 0.532 | 0.515 | 0.436 | 0.431 | 0.421 | 0.408 | 0.415 | 0.415 | 0.445 | 0.459 |
| | 720 | **0.448** | **0.438** | 0.454 | 0.441 | 0.491 | 0.459 | 0.462 | 0.449 | 0.478 | 0.450 | 0.666 | 0.589 | 0.489 | 0.462 | 0.481 | 0.441 | 0.473 | 0.451 | 0.543 | 0.490 |
| | Avg | **0.377** | **0.392** | 0.381 | 0.395 | 0.407 | 0.410 | 0.406 | 0.407 | 0.400 | 0.406 | 0.513 | 0.495 | 0.423 | 0.422 | 0.411 | 0.402 | 0.404 | 0.408 | 0.448 | 0.452 |
| ETTm2 | 96 | **0.174** | **0.256** | 0.175 | 0.258 | 0.180 | 0.264 | 0.183 | 0.270 | 0.187 | 0.267 | 0.287 | 0.366 | 0.197 | 0.296 | 0.183 | 0.266 | 0.193 | 0.293 | 0.203 | 0.287 |
| | 192 | **0.233** | **0.295** | 0.237 | 0.299 | 0.250 | 0.309 | 0.255 | 0.314 | 0.249 | 0.309 | 0.414 | 0.492 | 0.284 | 0.361 | 0.248 | 0.305 | 0.284 | 0.361 | 0.269 | 0.328 |
| | 336 | **0.294** | **0.333** | 0.298 | 0.340 | 0.311 | 0.348 | 0.309 | 0.347 | 0.321 | 0.351 | 0.597 | 0.542 | 0.381 | 0.429 | 0.309 | 0.343 | 0.382 | 0.429 | 0.325 | 0.366 |
| | 720 | **0.389** | **0.390** | 0.391 | 0.396 | 0.412 | 0.407 | 0.412 | 0.404 | 0.408 | 0.403 | 1.730 | 1.042 | 0.549 | 0.522 | 0.410 | 0.400 | 0.558 | 0.525 | 0.421 | 0.415 |
| | Avg | **0.272** | **0.318** | 0.275 | 0.323 | 0.288 | 0.332 | 0.290 | 0.334 | 0.291 | 0.333 | 0.757 | 0.610 | 0.353 | 0.402 | 0.287 | 0.329 | 0.354 | 0.402 | 0.305 | 0.349 |

---

**Algorithm 1** TimeCAT - Context-Aware Transformer Architecture.

---

**Require:** Input time series $\mathbf{X} \in \mathbb{R}^{T \times N}$; input length $T$; forecast horizon $S$; number of variables $N$; patch length $L_p$; stride $s$; number of groups $G$; embedding dimension $d$.

1: ▷ Instance normalization of the input series.
2: $\tilde{\mathbf{X}} = \texttt{InstanceNorm}(\mathbf{X})$ $\qquad\qquad\qquad\qquad\qquad\qquad$ ▷ $\tilde{\mathbf{X}} \in \mathbb{R}^{T \times N}$
3: ▷ Patch embedding: split series into overlapping patches.
4: $\mathbf{E}_n = \texttt{MLP}(\tilde{\mathbf{X}})$ $\qquad\qquad\qquad\qquad$ ▷ Patch embeddings $\mathbf{E}_n \in \mathbb{R}^{P \times d}$ for each variable
5: $\mathbf{E}_n = \mathbf{E}_n + \texttt{PE}$ $\qquad\qquad\qquad\qquad\qquad\qquad$ ▷ Add positional embeddings (PE)
6: ▷ Dynamic Grouping Layer: split series into $G$ groups.
7: $\mathbf{r} = \texttt{softmax}(\mathbf{W}_g \cdot \texttt{flatten}(\tilde{\mathbf{X}}') + \mathbf{b}_g)$ $\qquad\qquad\qquad$ ▷ Calculate group ratios
8: ▷ Token-Grouping-and-Merging: prepare new tokens by grouping.
9: $\mathbf{S}_n = [\mathbf{g}_n; \mathbf{g}_{n,1}; \ldots; \mathbf{g}_{n,G}; \mathbf{E}_n]$ $\qquad$ ▷ Append global and group tokens with length embedding
10: ▷ Context-Aware Mixing Block: process tokens for context mixing.
11: **for** $l = 1$ **to** $L$ **do**
12: $\qquad$ **Intra-Group Mixing:** Concatenate group and value tokens.
13: $\qquad \tilde{x} = \texttt{Self-Attention}([x_{\text{group}}, x_{\text{value}}])$
14: $\qquad \hat{x}_{\text{group}} = \texttt{LayerNorm}(x_{\text{group}} + \tilde{x})$
15: $\qquad \hat{x}_{\text{value}} = \texttt{LayerNorm}(x_{\text{value}} + \tilde{x})$
16: $\qquad$ **Inter-Group and Global Mixing:**
17: $\qquad Y_{\text{group}} = \texttt{MLP}_{\text{group}}(\hat{X}_{\text{group}}^{\top})$
18: $\qquad \tilde{X}_{\text{group}} = \texttt{LayerNorm}(\hat{X}_{\text{group}} + Y_{\text{group}}^{\top})$
19: $\qquad$ Update global token: $x_{\text{global}}^{\text{updated}} = x_{\text{global}} + (x_{\text{global, est}} - \texttt{Detach}(x_{\text{global, est}}))$
20: $\qquad y_{\text{global}} = \texttt{MLP}_{\text{global}}(x_{\text{global}}^{\text{updated}})$
21: $\qquad \tilde{x}_{\text{global}} = \texttt{LayerNorm}(x_{\text{global}}^{\text{updated}} + y_{\text{global}}) + \alpha * \text{Detach}(x_{global,est})$
22: **end for**
23: ▷ Context-Aware Sequence Reconstruction and Prediction.
24: $\mathbf{S} = [\tilde{x}_{\text{global}}, \tilde{X}_{\text{group}}, \hat{X}_{\text{value}}]$
25: $\hat{\mathbf{Y}} = \texttt{Predictor}(\mathbf{S})$ $\qquad\qquad\qquad$ ▷ Flatten, MLP, and de-normalize for final prediction
26: **Return** $\hat{\mathbf{Y}}$ $\qquad\qquad\qquad\qquad\qquad\qquad$ ▷ Return the forecast result $\hat{\mathbf{Y}} \in \mathbb{R}^{S \times N}$

---

outlined in the method. Consequently, the total computational complexity across all groups becomes:

$$\mathcal{O}_{\text{group}} = G \cdot \left(\frac{P}{G} + 1\right)^2 \cdot d$$

The computation reduction per variable, $\mathcal{R}$, is calculated as the relative decrease in complexity due to grouping:

$$\mathcal{R} = 1 - \frac{\mathcal{O}_{\text{group}}}{\mathcal{O}_{\text{no\_group}}} = 1 - \left(\frac{1}{G} + \frac{2}{P} + \frac{G}{P^2}\right).$$

In the typical case where $P \gg G$, the computation reduction $\mathcal{R}$ primarily depends on the number of groups $G$. Increasing $G$ generally results in a higher reduction rate in computational complexity.

The information exchange between group and global tokens is implemented using the MLP-Mixer mechanism, which is more efficient than self-attention modules, as demonstrated in the next experiment section. This further reduces the overall computational overhead. The reduction in computational overhead allows the model to handle longer sequences and larger datasets more efficiently, e.g., the foundation models.

### A.3.2 NEW TOKEN MECHANISM FOR TIME SERIES FOUNDATION MODELS

In the realm of time series forecasting, effectively capturing both global and local temporal dependencies is paramount for enhancing predictive accuracy and model robustness. Motivated by the intricate and hierarchical nature of time series data, we introduce a novel token mechanism within TimeCAT that distinguishes between global and group tokens. This mechanism leverages a dynamic grouping strategy to segregate variables into meaningful clusters, allowing the model to focus on high-level interactions through global tokens while simultaneously modeling detailed local patterns within each group. The intuition behind this approach is to mimic the hierarchical structure of temporal data, where overarching trends are captured by global tokens and finer-grained

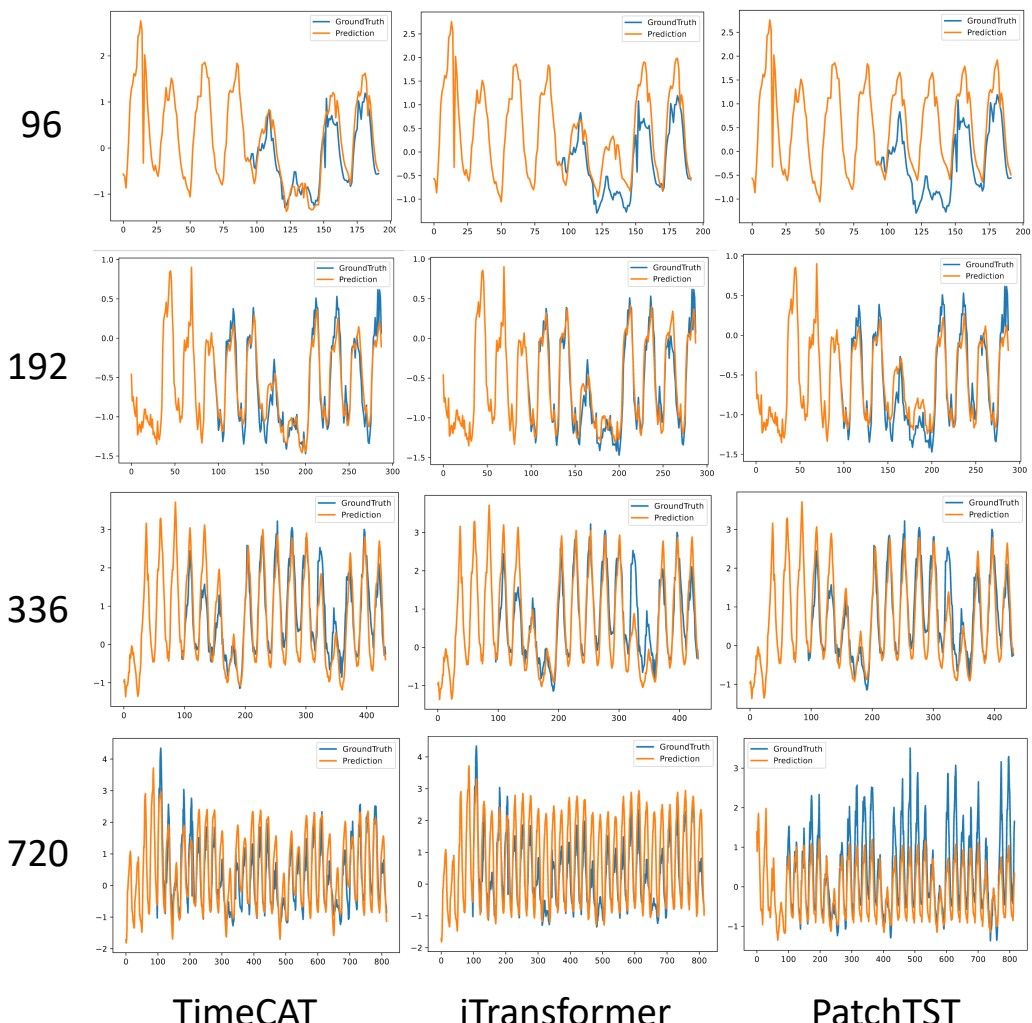

Figure 6: Visualization of ECL predictions of TimeCAT iTransformer and PatchTST over 96, 192, 336, 720 steps.

fluctuations are addressed by group tokens. By integrating skip connections between global and group tokens, our mechanism facilitates seamless information flow, enabling the model to learn complex and interdependent relationships across different temporal scales.

The benefits of this new token mechanism are multifaceted. Firstly, it enhances the model's ability to generalize across diverse datasets by providing a structured representation that encapsulates both broad and specific temporal dynamics. Secondly, the separation of global and group tokens reduces computational complexity by allowing parallel processing within groups, thereby improving efficiency without compromising accuracy. Additionally, this hierarchical token structure fosters interpretability, as it becomes easier to analyze and understand the contributions of global patterns versus localized trends in the forecasting process. Empirical results, as demonstrated in Table 5, validate that TimeCAT consistently outperforms existing state-of-the-art models, underscoring the effectiveness of our token mechanism in capturing the nuanced temporal relationships inherent in time series data. This advancement paves the way for more sophisticated and scalable foundation models in time series analysis, offering a robust framework for future research and applications.

## A.4 VISUALIZATION OF PREDICTION RESULTS

Figure 6 highlights the superior performance of TimeCAT in forecasting electricity consumption levels (ECL) across various steps (96, 192, 336, 720). Compared to iTransformer and PatchTST, TimeCAT consistently exhibits closer alignment with the ground truth, especially noticeable in the long-term forecasts of 336 and 720 steps, demonstrating its robust predictive capability and reliability in handling complex time series data.

