# OpenReview forum: "TimeCAT: Hierarchical Context-Aware Transformer with Dynamic Grouping for Time Series Forecasting"
_ICLR.cc/2025/Conference — ICLR 2025 Conference Withdrawn Submission_

### Official Review · Reviewer_h2kY · 2024-11-02

**Soundness:** 3
**Presentation:** 2
**Contribution:** 3
**Rating:** 3
**Confidence:** 3

**Summary:**

This manuscript introduces a framework that employs a hierarchical context-aware transformer, TimeCAT, designed to dynamically group time series patches and efficiently capture dependencies. TimeCAT was compared with recent state-of-the-art methods, and the results demonstrate promising accuracy.

**Strengths:**

The proposed TimeCAT model was evaluated against benchmarks, and its accuracy shows promising results compared to recent state-of-the-art baselines. The Context-Aware Mixing Block enhances information exchange. In particular, the grouping mechanism allows the transformer to capture dependencies between highly related patches, thereby eliminating unnecessary computations between less related patches and improving efficiency. This mechanism demonstrates potential for effectively extracting information from Multivariate Time Series (MTS) data.

**Weaknesses:**

The hyperparameter path size, \(P\), is crucial for capturing temporal dependencies and reducing computational complexity. This manuscript notes that in scenarios where \(P\) is significantly greater than \(G\), increasing \(G\) leads to higher computational savings. However, a detailed discussion on the size of \(P\) is absent. A parameter sensitivity study regarding \(P\), along with a table similar to Table 3 to highlight the computational savings, would strengthen the manuscript. Additionally, providing a specific example that compares the complexity reduction in terms of parameters, FLOPs, and memory would be beneficial.

The clarity of the context-aware mixing workflow could be improved by detailing the dimensions of all intermediate tensors.

While the efficiency of TimeCAT is highlighted as a key contribution, the manuscript does not compare it with other efficient transformer-based MTS forecasting baselines. This comparison would provide a more comprehensive evaluation of TimeCAT's performance and efficiency.

**Questions:**

I have a few suggestions which may improve the quality of the paper:
(a) Please revise Section 3 to enhance its readability.
(b) Include a qualitative comparison to demonstrate improvements in computational complexity in terms of parameters, FLOPs, and memory.
(c) I recommend including transformer-based baselines that aim to improve efficiency.

---

### Official Review · Reviewer_RrDg · 2024-11-03

**Soundness:** 3
**Presentation:** 2
**Contribution:** 3
**Rating:** 5
**Confidence:** 4

**Summary:**

TimeCAT is designed to address the limitations of existing Transformer-based models that struggle with capturing complex temporal dependencies and suffer from computational inefficiencies, particularly with long sequences. The core innovation of TimeCAT is its dynamic grouping mechanism, which segments input sequences into semantically coherent groups, enabling efficient hierarchical mixing at different levels of context. This approach facilitates the modeling of both local patterns within groups and global trends across the entire sequence.

**Strengths:**

1. TimeCAT's hierarchical structure, with its focus on intra-group, inter-group, and global interactions, provides a comprehensive framework for capturing both local and global temporal patterns. This multi-scale modeling is a significant advancement in the field.
2. The paper demonstrates a substantial reduction in computational complexity by applying self-attention within groups rather than across the entire sequence. This efficiency gain is crucial for handling longer sequences and larger datasets.

**Weaknesses:**

1. The main experiments in the article are limited. It might be worth considering adding short-term experiments and incorporating new datasets. For example, there are many new datasets available here: https://huggingface.co/datasets/Salesforce/lotsa_data.
2. How does the model’s performance change as the input length increases?
3. Wouldn’t using the downsampled version of x to perform group division result in information loss?
4. The spacing between the image and the title is too large.

**Questions:**

1. The writing of the article needs improvement. In Equation 3, what does g' mean?

---

### Official Review · Reviewer_gtQK · 2024-11-06

**Soundness:** 1
**Presentation:** 1
**Contribution:** 2
**Rating:** 3
**Confidence:** 4

**Summary:**

This paper proposes TimeCAT, a transformer-based model for time-series forecasting that leverages hierarchical dependencies among fixed-length patches, dynamically grouped patches, and a global token. Specifically, TimeCAT first forms dynamic groups of input patches, then captures both fine-grained and coarse-grained information through hierarchical dependencies using Context-Aware Mixing Blocks. Experimental results demonstrate that TimeCAT outperforms previous models on several time-series forecasting benchmarks.

**Strengths:**

1. This paper is well-motivated.
2. The proposed model, TimeCAT, outperforms prior models in several time-series forecasting benchmarks.

**Weaknesses:**

1. The presentation quality of this paper is unsatisfactory.
   - The method is not clearly described throughout the paper. The descriptions are overly wordy, with many unnecessary and confusing notations.
   - For instance, from group ratios $\mathbf{r} \in \mathbb{R}^G$, how are group indices obtained, and how are they optimized? Such indices are often assigned through discretization, making it unclear how they can be optimized using gradient-based methods. In particular, it is unclear how to compute gradients for $\mathbf{W}\_g$ and $\mathbf{b}\_g$. Additionally, as group size is calculated using the ceiling function, the optimization of the embedding $\mathbf{l}\_{s_i}$ is also unclear.
   - Other confusing notations are as follows: What are $\mathbf{g}'\_{n,i}$ in Eq(3) and $\tilde{\mathbf{g}}\_{n,i}$ in Eq (7)? Is $RD$ a single hyperparameter or a product of two hyperparameters, $R$ and $D$?
   - Notational consistency is also an issue, with symbols such as $\textbf{X}$ vs $X$ and $x$ vs $\tilde{x}$ vs $\hat{x}$. The frequent use of accents and subscripts/superscripts could easily confuse readers.
2. Claims are not well supported.
   - The paper emphasizes the importance of the dynamic grouping mechanism. However, the experiments show that $G=2$ is sufficient to achieve good results. This very small number of groups does not adequately demonstrate the necessity of grouping for solving the problem, as all groups may still be too coarse.
   - Additionally, the grouping is determined by a single linear transformation of the input matrix, which raises doubts about the quality of the grouping.
   - Why does training become unstable without Eq (12) and (15)? Furthermore, there is no ablation study to justify the inclusion of this gradient detachment technique.
   - What is the rationale for the order of operations in the context-aware mixing block? One could simply apply self-attention across all tokens to capture intra-group and inter-group dependencies. If $G$ is as small as used in this paper (e.g., $G=2$), the computational complexity is not significantly greater than that of TimeCAT. The hierarchical design lacks both a clear explanation and experimental validation.
   - Why does Figure 5 highlight the effectiveness of the grouping strategy? For example, I cannot see any alignment between Figures 5(a) and 5(b), and the t-SNE plots in Figures 5(c)-(e) show no meaningful pattern. Why should distinct clusters of global tokens reflect effective separation of variables and high-level interactions? A more detailed explanation would be helpful.

**Questions:**

- How is $\tilde{X}$ downsampled to obtain $\tilde{X}'$?
- Time-series forecasting experiments are often sensitive to hyperparameters, such as learning rates and batch sizes. How were these parameters chosen?
- Could you provide a comparison of the actual training times for the models listed in Table 2?
- Could you include results for cases when $G=1$ and when $G$ is large (e.g., $G=16$, $G=32$)?
- There are many unnecessary horizontal spaces between equations and sentences (e.g., L222, L226, and L301). These should be removed.

---

### Note · Authors · 2024-11-12

I have read and agree with the venue's withdrawal policy on behalf of myself and my co-authors.